## [Decision Letter · Decision Letter 0]

10 Nov 2023

PONE-D-23-30534The role of sociodemographic factors on the acceptability of digital mental health care:

A scoping review protocolPLOS ONE

Dear Dr. Lal,

Thank you for submitting your manuscript to PLOS ONE. After careful consideration, we feel that it has merit but does not fully meet PLOS ONE’s publication criteria as it currently stands. Therefore, we invite you to submit a revised version of the manuscript that addresses the points raised during the review process.

We look forward to receiving your revised manuscript.

Kind regards,

Maher Abdelraheim Titi

Academic Editor

PLOS ONE

Journal Requirements:

SL is an Associate Professor at the University of Montreal and leads a research program in the field of digital mental health. In the past 5 years SL has received research funding from the Canadian Institutes of Health Research, Canada Research Chairs program, Hoffman-La Roche, Foundation of Stars to advance this work. All of these are unrelated to this specific study. 

NA is a graduate student at the University of Montreal and is conducting this work towards partial fulfillment of requirements for a Masters of Science degree under the supervision of SL.

Reviewers' comments:

Reviewer's Responses to Questions

**Comments to the Author**

1. Does the manuscript provide a valid rationale for the proposed study, with clearly identified and justified research questions?

Reviewer #1: Yes

Reviewer #2: Partly

Reviewer #3: Yes

2. Is the protocol technically sound and planned in a manner that will lead to a meaningful outcome and allow testing the stated hypotheses?

Reviewer #1: Yes

Reviewer #2: Partly

Reviewer #3: Yes

3. Is the methodology feasible and described in sufficient detail to allow the work to be replicable?

Reviewer #1: Yes

Reviewer #2: No

Reviewer #3: Yes

4. Have the authors described where all data underlying the findings will be made available when the study is complete?

Reviewer #1: Yes

Reviewer #2: Yes

Reviewer #3: Yes

5. Is the manuscript presented in an intelligible fashion and written in standard English?

Reviewer #1: Yes

Reviewer #2: Yes

Reviewer #3: Yes

6. Review Comments to the Author

You may also provide optional suggestions and comments to authors that they might find helpful in planning their study.

Reviewer #1: **Summary**

This paper proposes a protocol for a scoping review that synthesizes findings across the literature related to socio-demographic variables that impact acceptability of digital mental health innovations (DMHIs).

The manuscript proposes a worthwhile review and is overall straightforward to read. I also appreciate the transparent and extensive description of the proposed search strategy! I have a number of comments regarding details that could benefit from clarification, which I have listed below.

**Comments**

1. Digital mental health innovations is abbreviated to DMHIs in the abstract but nowhere else in the paper.

2. The terms digital mental health innovations, digital mental health care and digital mental health interventions are used interchangeably. Do these refer to the same or different things?

3. Based on the introduction of the paper, it seems that the main research question is to identify what sociodemographic factors affect perceived acceptability. However, I don’t see this listed as a research question.

4. A broad definition of acceptability is included in the Introduction. Will any study considering acceptability be included, no matter how they measured/defined acceptability? For example, there are validated survey scales assessing acceptability, but acceptability can also be studied in a more qualitative way through interviews.

5. Are there any restrictions on study method or will any study considering acceptability (e.g. both quantitative and qualitative studies) be considered?

6. L50: can the authors give any examples of existing barriers to access mental health services?

7. L59-78 is a bit repetitive describing one review at a time. It may improve flow to succinctly summarize what other reviews have done, what they have not done, and how this review will bridge that gap.

8. L113: can you be more specific than ‘which digital mental health innovations’? How are you going to describe and/or categorize them? By type of platform (e.g. website, app), type of mental health symptoms it intends to address (e.g. anxiety, depression), type of population it focuses on, type of care it offers, etc.

9. Can the authors clarify the inclusion criterion that 90% or more of the study sample needs to represent a specific sociodemographic group? If 60% of study participants identify as men, but the study showed that male participants rated digital mental health interventions significantly higher on acceptability than other gender identities, would this study not be relevant to the review?

10. It would help with readability to include an overview of all inclusion and exclusion criteria in a bullet-point list or table.

11. L215-216: how will the grey literature be used? Will theses and dissertations be included in the review as articles, or will the reference lists of theses be used to identify other peer-reviewed studies?

12. L329: how are absolute findings evaluated? E.g. for the example that is given, ‘men demonstrated a high level of perceived acceptability’ => this may vary across the articles you are including in your review, but how is ‘high’ determined? Are there validated cut-off scores that determine whether something has a high score of acceptability? This goes back to my earlier comment whether only a quantitative measure of acceptability will be considered or if other assessments of acceptability will be included in the review as well.

13. L343: it is mentioned that the review will not evaluate the methodological rigor of the included studies. However, will some kind of other, less rigorous, quality assessment be carried out to screen articles?

Reviewer #2: General comments

I read with interest your scoping review protocol on the role of sociodemographic factors on the acceptability of digital mental health care. This review will add an important contribution to the eMental health field. Below are some suggestions to improve the clarity, and add to the quality, of this proposal. I wish you all the best with your work!

Major inquiries

1. It is unclear to me how research question 1 (page 5, line 113) directly addresses your objective on what is the role of sociodemographics in the degree of perceived acceptability of DMHI (lines 107-108). Can you please clarify?

Of note, while you should describe the sample of studies of DMHI included in the review, that does not mean this summary should form a research question. (Also, if you answer question 3, you will be indicating which interventions have been evaluated through a sociodemographic lens.)

2. Research question 3 (page 6, lines 117-118) does not include the neutral category that you outline in the Data Synthesis section (page 15, line 326). If your objective is focused on the degree of perceived acceptability, it seems that ‘neutral’ would be important to acknowledge as part of the spectrum?

3. Determining study inclusion necessitates a detailed evaluation—1) determining the report of relative or absolute acceptability, and if absolute acceptability is reported, the need to determine if at least 90% of the sample represents a specific sociodemographic group; and 2) determining if acceptability is assessed independently in the case of blended interventions. Given these requirements for inclusion, it is important to include details on the screening strategy (tasks/steps; page 12) and any instructions/parameters for determining these criteria (e.g., Which values are used for calculating the proportion of the study sample? How to determine an independent acceptability assessment – Can multivariable modelling results be used? Does the acceptability finding need to be reported on its own?). The inclusion of these details will further operationalize the review and promote transparency and replicability.

4. Page 15, line 332: It is stated that additional findings may be reported based on the identification of patterns. What analytic approach will be used to identify patterns?

5. Page 15, line 334: You note that it may be necessary to use visual representation if the findings are complex. Can you describe more details regarding ‘complexity’ to promote transparency and replicability? In particular, how are you defining complexity, and what is the process that will be used to determine complexity?

6. I am curious to know more about your approach to defining high acceptability (report in a study of a positive association) (page 15, line 326). Typically, general interpretation of measures of association follows this guide: weak (0.01-0.09), moderate (0.10-0.29), strong (0.30-0.59), very strong (0.60-0.99). Based on your definition, in your review, it reads that you will report a weak, positive association as reflecting high acceptability. Can you share your thoughts on whether or not you think your interpretation of any positive association as ‘high acceptability’ could be misleading, and why/why not?

7. In its present state, the conclusion serves as a detailed account of the strengths and limitations of the review. These should be stated in its own section. The conclusion should serve as a brief summary. Revisions are necessary.

Minor inquiries and comments

1. I understand that you are interested in synthesizing studies of associations between sociodemographics and perceived acceptability. As this is a scoping review, would you consider adding another objective to slightly broaden the focus? In addition to reviewing studies that examined associations, would you also consider reviewing all studies that included examining perceived acceptability of a DMHI. I believe these studies would be included in the search strategy. For this group of studies, you could review the participant characteristics across studies to explore the diversity from which acceptability findings are based on. While this addition broadens the review, it would add value as you could comment on the diversity of study samples and what is needed moving forward in terms of EDI in the DMHI research field.

2. In the introduction, the second paragraph is repetitive in structure: sentence introducing a prior review followed by a sentence starting with ‘however’. This paragraph requires revision to improve reader engagement.

3. On page 5 in the statement of the primary objective, you can remove the examples in brackets “(e.g., age, ethnicity, gender)”. Objectives do not need parenthesized examples as operational details should be described in the methods. This is what you have done on the following page under Population.

4. Page 10, line 213: “Only materials published between January 2013 and June 2023 will be considered.” This is already stated on the previous page (page 9; line 199). Similarly, the content on page 10, line 214 is also duplicative “Moreover, the reference list of secondary sources, identified as relevant to the topic of inquiry, will be examined.” See page 9, lines 202 and 203.

5. Page 14, line 303 should be “As data are extracted” (plural); line 315 should be “The extracted data will be” as extraction is a descriptive word in this sentence.

6. Page 14, line 315. I suggest integrating this sentence where it is relevant as it is not a stand-alone paragraph. (I suspect you can move it to the paragraph starting on line 324, re: summary of the key findings on the role of sociodemographics?)

Reviewer #3: An appropriate and timely protocol proposing to synthesise data on the experience, specifically the acceptability, of taking part in digital health interventions, amongst persons who have taken part in such interventions focused on mental health outcomes. Acceptability as a concept is clearly operationalised. The literature search and analysis approach seem appropriate and the inclusion and focus on demographic sub-group analysis is very important and will ensure the results are meaningful and again there are clear criteria for inclusion or exclusion of studies based on the way the demographic data are presented. Overall, this is a well-written and competently described review protocol. One minor comment - the focus in the opening line on Canada is not necessary or appropriate for an international journal.

7. PLOS authors have the option to publish the peer review history of their article (what does this mean?). If published, this will include your full peer review and any attached files.

Reviewer #1: **Yes: **Judith Borghouts

Reviewer #2: No

Reviewer #3: **Yes: **Prof. Brian McGuire

---

## [Author Response · Author response to Decision Letter 0]

24 Jan 2024

Below are the comments we have received from the reviewers and our responses

Reviewer 1 Comments:

1. Digital mental health innovations is abbreviated to DMHIs in the abstract but nowhere else in the paper

Our response: We made revisions to ensure the abbreviation is used throughout the manuscript. 

2. The terms digital mental health innovations, digital mental health care and digital mental health interventions are used interchangeably. Do these refer to the same or different things?

Our response: We used all of these terms interchangeably, however, for greater clarity, we decided to use digital mental health innovations more consistently throughout the manuscript, and we have used digital mental health care only a few times in cases where we felt that it allowed for better flow. 

3. Can the authors give any examples of existing barriers to access mental health services?

Our response: We have provided examples of barriers to access to mental health services (e.g., limited availability of services).

4. L59-78 is a bit repetitive describing one review at a time. It may improve flow to succinctly summarize what other reviews have done, what they have not done, and how this review will bridge that gap

Our response: We have revised this section to be more succinct. 

5. Based on the introduction, it seems that the main research question is to identify what sociodemographic factors affect perceived acceptability. However, I don’t see this listed as a research question.

Our response: We have added the following question “Which sociodemographic factors have been positively, negatively, or not associated with the perceived acceptability of DMHI?”

6. L113: can you be more specific than ‘which digital mental health innovations’? How are you going to describe and/or categorize them? By type of platform (e.g. website, app), type of mental health symptoms it intends to address (e.g. anxiety, depression), type of population it focuses on, type of care it offers, etc.

Our response: We have provided more information on how the different innovations will be described. Specifically, we have added the following statement “Please see the Study Selection and Extraction section below for more details on how the various DMHI will be categorized”. 

7. A broad definition of acceptability is included in the Introduction. Will any study considering acceptability be included, no matter how they measured/defined acceptability? For example, there are validated survey scales assessing acceptability, but acceptability can also be studied in a more qualitative way through interviews.

Our response: We have clarified as part of the eligibility criteria that acceptability will need to be evaluated through quantitative methods. 

8. Are there any restrictions on study method or will any study considering acceptability (e.g. both quantitative and qualitative studies) be considered?

Our response: Studies using a qualitative design will be excluded (This has been clarified in the manuscript). We will only consider studies with a quantitative or mixed-methods design (for the quantitative component of those studies). 

9. Can the authors clarify the inclusion criterion that 90% or more of the study sample needs to represent a specific sociodemographic group? If 60% of study participants identify as men, but the study showed that male participants rated digital mental health interventions significantly higher on acceptability than other gender identities, would this study not be relevant to the review?

Our response: Yes, these results would be pertinent for the proposed review. We have modified the eligibility criteria in our manuscript. We will only consider studies in which relative acceptability is reported (e.g., Men have higher perceived acceptability than women), the absolute acceptability of a specific group in comparison to another is reported (e.g., The mean acceptability score of men is 4 and the mean acceptability score of women is 3.5) or in which the study sample is homogeneous for at least one sociodemographic factor of interest (e.g., every study participant identifies as a woman). 

10. It would help with readability to include an overview of all inclusion and exclusion criteria in a bullet-point list or table.

Our response: The inclusion and exclusion criteria of the proposed review will be added as a table in the revised manuscript.

11. How will the grey literature be used? Will theses and dissertations be included in the review as articles, or will the reference lists of theses be used to identify other peer-reviewed studies?

Our response: As part of our revisions, we have established, based on a preliminary scanning of the literature, that we will not consider non-peer-reviewed studies for inclusion, including theses and dissertations. However, the reference list of peer-reviewed secondary sources will be examined to identify articles that may be eligible for inclusion in the proposed review. 

12. *L329: how are absolute findings evaluated? E.g. for the example that is given, ‘men demonstrated a high level of perceived acceptability’ => this may vary across the articles you are including in your review. How is ‘high’ determined? Are there validated cut-off scores that determine whether something has a high score of acceptability? 

Our response: We have clarified in the manuscript that we will finalize our approach depending on the way acceptability is measured across included studies. However, as of now, we have added the following clarification: “For absolute acceptability, the cut-off scores reported in the literature for the various measures used to measure acceptability across studies will be used to indicate whether the reported scores represent high or low acceptability. If such scores are not available, the acceptability scores that have been evaluated using the same measures across studies will be compared. For studies that utilized non-standardized measures to examine the perceived acceptability of users, the interpretation of these scores documented in the studies will be reported in the proposed review.”

13. L343: it is mentioned that the review will not evaluate the methodological rigor of the included studies. However, will some kind of other, less rigorous, quality assessment be carried out to screen articles?

Our response: At the moment, we are not planning on conducting a quality assessment of included articles. However, we will report on the methodological design of each article that is included in the proposed review (e.g., RCT, Case study) to help readers in their interpretation of the results. Quality assessments are typically not required in scoping review methodology.

Reviewer 2 Comments

1. Page 10, line 213: “Only materials published between January 2013 and June 2023 will be considered.” This is already stated on the previous page (page 9; line 199). Similarly, the content on page 10, line 214 is also duplicative “Moreover, the reference list of secondary sources, identified as relevant to the topic of inquiry, will be examined.” See page 9, lines 202 and 203.

Our response: We have removed the duplicated information from the Sources of Evidence section. 

2. Page 14, line 303 should be “As data are extracted” (plural); line 315 should be “The extracted data will be” as extraction is a descriptive word in this sentence.

Our response: This correction has been reflected in the manuscript. 

3. Page 14, line 315. I suggest integrating this sentence where it is relevant as it is not a stand-alone paragraph. (I suspect you can move it to the paragraph starting on line 324, re: summary of the key findings on the role of sociodemographics?)

Our response: This suggestion has been reflected in the manuscript. 

4. In the introduction, the second paragraph is repetitive in structure: sentence introducing a prior review followed by a sentence starting with ‘however’. This paragraph requires revision to improve reader engagement.

Our response: We have revised this section to improve the flow.

5. It is unclear to me how research question 1 (page 5, line 113) directly addresses your objective on what is the role of sociodemographics in the degree of perceived acceptability of DMHI (lines 107-108). Can you please clarify?

Our response: We have removed question 1 "Which digital mental health innovations have had their perceived acceptability evaluated through a consideration of sociodemographic factors?" However, as suggested by reviewer 2, this information will be reported in a table.

6. Research question 3 (page 6, lines 117-118) does not include the neutral category that you outline in the Data Synthesis section (page 15, line 326). If your objective is focused on the degree of perceived acceptability, it seems that ‘neutral’ would be important to acknowledge as part of the spectrum?

Our response: This question has been reformulated to the following “Which sociodemographic factors have been positively, negatively, or not associated with the perceived acceptability of DMHI?”

7. I understand that you are interested in synthesizing studies of associations between sociodemographics and perceived acceptability. As this is a scoping review, would you consider adding another objective to slightly broaden the focus? In addition to reviewing studies that examined associations, would you also consider reviewing all studies that included examining perceived acceptability of a DMHI. I believe these studies would be included in the search strategy. For this group of studies, you could review the participant characteristics across studies to explore the diversity from which acceptability findings are based on. While this addition broadens the review, it would add value as you could comment on the diversity of study samples and what is needed moving forward in terms of EDI in the DMHI research field.

Our response: Absolutely, this has been the plan, but we can understand how this may not have been reflected in our first iteration of the manuscript. We have clarified in the introduction what we mean by association. Specifically, we have added the following: “In the context of this review, the term “association” extends beyond its statistical sense and refers to the presence of qualitative patterns of association between specific sociodemographic groups within and across studies, and their perceived acceptability of DMHI.”

8. On page 5 in the statement of the primary objective, you can remove the examples in brackets “(e.g., age, ethnicity, gender)”. Objectives do not need parenthesized examples as operational details should be described in the methods. This is what you have done on the following page under Population.

Our response: We have removed the parenthesized examples from this section. 

9. *Determining study inclusion necessitates a detailed evaluation—1) determining the report of relative or absolute acceptability, and if absolute acceptability is reported, the need to determine if at least 90% of the sample represents a specific sociodemographic group; and 2) determining if acceptability is assessed independently in the case of blended interventions. Given these requirements for inclusion, it is important to include details on the screening strategy and any instructions/parameters for determining these criteria (e.g., Which values are used for calculating the proportion of the study sample? How to determine an independent acceptability assessment – Can multivariable modelling results be used? Does the acceptability finding need to be reported on its own?). The inclusion of these details will further operationalize the review and promote transparency and replicability.

Our response: As part of our revisions, we have modified the eligibility criteria in our manuscript. We will only consider studies in which relative acceptability is reported (e.g., Men have higher perceived acceptability than women), the absolute acceptability of a specific group in comparison to another is reported (e.g., The mean acceptability score of men is 4, and the mean acceptability score of women is 3.5) or in which the study sample is homogeneous for at least one sociodemographic factor of interest (e.g., every study participant identifies as a woman). Therefore, the criterion of 90% is no longer relevant for the proposed review. For the second component of your comment on how we will determine if acceptability is assessed independently in blended interventions, we will examine whether the perceived acceptability score for a blended intervention (e.g., In-person psychotherapy and ICBT) is reported as a global score or if the acceptability of the ICBT component was evaluated separately. If it has not been evaluated separately, we would not include the study in the proposed review. All of these clarifications have been reflected in the new iteration of the manuscript. 

10. Page 15, line 332: It is stated that additional findings may be reported based on the identification of patterns. What analytic approach will be used to identify patterns?

Our response: This statement specifically refers to the possibility of additional categories of information emerging as pertinent (when the data extraction strategy is piloted) that were not considered in this protocol. 

11. Page 15, line 334: You note that it may be necessary to use visual representation if the findings are complex. Can you describe more details regarding ‘complexity’ to promote transparency and replicability? In particular, how are you defining complexity, and what is the process that will be used to determine complexity?

Our response: Complexity here refers to the quantity of data that will be presented to readers. If we find that there is a large amount of data to be extracted for a particular category of information, it may be more accessible for readers to have visual representations of the data to enhance their understanding of our findings. This clarification has been reflected in the new iteration of the manuscript. 

12. I am curious to know more about your approach to defining high acceptability (page 15, line 326). Typically, general interpretation of measures of association follows this guide: weak (0.01-0.09), moderate (0.10-0.29), strong (0.30-0.59), very strong (0.60-0.99). Based on your definition, in your review, it reads that you will report a weak, positive association as reflecting high acceptability. Can you share your thoughts on whether or not you think your interpretation of any positive association as ‘high acceptability’ could be misleading, and why/why not?

Our response: In the new iteration of the manuscript, we have clarified what we mean by associations. Here, we will not attempt to interpret or calculate statistical associations, but these will be stated if they are reported by the authors. Instead, we are looking for qualitative patterns in our findings. For example, if we notice that individuals identifying as women have reported high acceptability in 7 studies, this would be reported. To clarify how we will determine whether acceptability is high or not, we have added the following statement to the manuscript: “For absolute acceptability, the cut-off scores reported in the literature for the various measures used to measure acceptability across studies will be used to indicate whether the reported scores represent high or low acceptability. If such scores are not available, the acceptability scores that have been evaluated using the same measures across studies will be compared. For studies that utilized non-standardized measures to examine the perceived acceptability of users, the study’s author’s interpretation of these scores will be reported in the proposed review.” Please note that we will finalize our approach depending on the way acceptability was measured across the included studies.

13. In its present state, the conclusion serves as a detailed account of the strengths and limitations of the review. These should be stated in its own section. The conclusion should serve as a brief summary.

Our response: In the new iteration of the manuscript, we have separated the original Conclusion section into Limitations and Conclusion. 

Reviewer’3 Comments:

1. One minor comment - the focus in the opening line on Canada is not necessary or appropriate for an international journal.

Our

---

## [Decision Letter · Decision Letter 1]

16 Feb 2024

PONE-D-23-30534R1The role of sociodemographic factors on the acceptability of digital mental health care:

A scoping review protocolPLOS ONE

Dear Dr. Lal,

Thank you for submitting your manuscript to PLOS ONE. After careful consideration, we feel that it has merit but does not fully meet PLOS ONE’s publication criteria as it currently stands. Therefore, we invite you to submit a revised version of the manuscript that addresses the points raised during the review process.

We look forward to receiving your revised manuscript.

Kind regards,

Maher Abdelraheim Titi

Academic Editor

PLOS ONE

Journal Requirements:

Reviewers' comments:

Reviewer's Responses to Questions

**Comments to the Author**

1. Does the manuscript provide a valid rationale for the proposed study, with clearly identified and justified research questions?

Reviewer #1: Yes

Reviewer #4: Yes

2. Is the protocol technically sound and planned in a manner that will lead to a meaningful outcome and allow testing the stated hypotheses?

Reviewer #1: Yes

Reviewer #4: Yes

3. Is the methodology feasible and described in sufficient detail to allow the work to be replicable?

Reviewer #1: Yes

Reviewer #4: Yes

4. Have the authors described where all data underlying the findings will be made available when the study is complete?

Reviewer #1: Yes

Reviewer #4: Yes

5. Is the manuscript presented in an intelligible fashion and written in standard English?

Reviewer #1: Yes

Reviewer #4: Yes

6. Review Comments to the Author

You may also provide optional suggestions and comments to authors that they might find helpful in planning their study.

Reviewer #1: I thank the authors for their responses to our comments and revision of the paper.

I believe the authors’ efforts to address our comments greatly improve the protocol, and I am happy with the changes made.

Reviewer #4: Thank you for the opportunity to review this paper as a third reviewer. I have seen the previous versions and comments from reviewers 1 and 2 and am glad the issues have been addressed.

I have one suggestion related to the concept of 'acceptability', which is well defined in the paper, but less so in the search strategy which was included as an appendix. The terms 'patient preference', 'consumer satisfaction' and 'patient satisfaction' have not been included, and it seems these very similar - yet admittedly different - concepts may also yield important results. I'm wondering if including these terms in the first lines of the search along with the acceptability terms (lines 1-4) would be useful. If in testing these terms, they are not of use, perhaps a short explanation could be added? I am not confident that other authors will have as well defined a concept of acceptability and will be using other definitions.

7. PLOS authors have the option to publish the peer review history of their article (what does this mean?). If published, this will include your full peer review and any attached files.

Reviewer #1: **Yes: **Judith Borghouts

Reviewer #4: **Yes: **Amanda Ross-White

---

## [Author Response · Author response to Decision Letter 1]

29 Feb 2024

We thank the reviewers for their constructive reviews and are encouraged to receive feedback that our revisions have “greatly improved the protocol”. We greatly appreciate the opportunity to submit a further revision of our paper for your continued consideration. Below, we respond to each reviewer’s comments and provide an overview of the corresponding tracked changes we made in the paper.

Reviewer 1 Comment:

1. I thank the authors for their responses to our comments and revision of the paper. I believe the authors’ efforts to address our comments greatly improve the protocol, and I am happy with the changes made.

Our response: We are happy to hear that the changes we have made are satisfactory. Thank you for the thoughtful feedback to help improve the manuscript.

Reviewer 2 Comments: 

1. I have one suggestion related to the concept of 'acceptability', which is well defined in the paper, but less so in the search strategy which was included as an appendix. The terms 'patient preference', 'consumer satisfaction' and 'patient satisfaction' have not been included, and it seems these very similar - yet admittedly different - concepts may also yield important results. I'm wondering if including these terms in the first lines of the search along with the acceptability terms (lines 1-4) would be useful. If in testing these terms, they are not of use, perhaps a short explanation could be added? I am not confident that other authors will have as well defined a concept of acceptability and will be using other definitions.

Our response: We appreciate this suggestion very much. This concern also emerged as we were in the process of developing the search strategies for the present review. Our decision to omit these terms is based on the following: 1) First, the conceptualization of acceptability in this protocol is informed by the Technology Acceptance Model (TAM) and its two core components, perceived ease of use and perceived utility. In contrast to general satisfaction and preference, the TAM suggests, through its two core components, specific criteria to consider as one evaluates a given innovation. Despite the possibility of variations in the conceptualization of acceptability across studies, the general criteria used to evaluate acceptability based on the TAM should generally remain the same; 2) Furthermore, this decision was made to limit the scope of this review. Indeed, as evidenced by our search strategy, we have tried to be as inclusive as possible for the population and context component of this review. However, to ensure the feasibility of this review, this decision also meant that the concept component of the review would need to be scoped down. Our current strategy retrieved 12,894 articles (with duplicates removed) that will need to be screened – demonstrating already a broad approach. All this being said, we can appreciate how these decisions may impact our results. Therefore, we have added in the “limitation” section of the manuscript that the omission of these terms represents a limitation of this review.

---

## [Decision Letter · Decision Letter 2]

25 Mar 2024

The role of sociodemographic factors on the acceptability of digital mental health care:

A scoping review protocol

PONE-D-23-30534R2

Dear Dr. Lal,

We’re pleased to inform you that your manuscript has been judged scientifically suitable for publication and will be formally accepted for publication once it meets all outstanding technical requirements.

Kind regards,

Maher Abdelraheim Titi

Academic Editor

PLOS ONE

Reviewers' comments:

Reviewer's Responses to Questions

**Comments to the Author**

1. Does the manuscript provide a valid rationale for the proposed study, with clearly identified and justified research questions?

Reviewer #4: Yes

2. Is the protocol technically sound and planned in a manner that will lead to a meaningful outcome and allow testing the stated hypotheses?

Reviewer #4: Yes

3. Is the methodology feasible and described in sufficient detail to allow the work to be replicable?

Reviewer #4: Yes

4. Have the authors described where all data underlying the findings will be made available when the study is complete?

Reviewer #4: Yes

5. Is the manuscript presented in an intelligible fashion and written in standard English?

Reviewer #4: Yes

6. Review Comments to the Author

You may also provide optional suggestions and comments to authors that they might find helpful in planning their study.

Reviewer #4: You have adequately explained the search rationale and addressed the limitations. Thank you for this revision.

7. PLOS authors have the option to publish the peer review history of their article (what does this mean?). If published, this will include your full peer review and any attached files.

Reviewer #4: **Yes: **Amanda Ross-White
